# Analysis of Patient Safety Incidents in Primary Care Reported in an Electronic Registry Application

**DOI:** 10.3390/ijerph18178941

**Published:** 2021-08-25

**Authors:** Montserrat Gens-Barberà, Núria Hernández-Vidal, Elisa Vidal-Esteve, Yolanda Mengíbar-García, Immaculada Hospital-Guardiola, Eva M. Oya-Girona, Ferran Bejarano-Romero, Carles Castro-Muniain, Eva M. Satué-Gracia, Cristina Rey-Reñones, Francisco M. Martín-Luján

**Affiliations:** 1Quality and Patient Safety Central Functional Unit, Gerència d’Atenció Primària Camp de Tarragona, Institut Català de la Salut, 43005 Tarragona, Spain; nhernandez.tgn.ics@gencat.cat (N.H.-V.); montsegens@gmail.com (E.V.-E.); ymengibar.tgn.ics@gencat.cat (Y.M.-G.); ihospitalg.tgn.ics@gencat.cat (I.H.-G.); eoya.tgn.ics@gencat.cat (E.M.O.-G.); fbejarano.tgn.ics@gencat.cat (F.B.-R.); ccastro.tgn.ics@gencat.cat (C.C.-M.); 2Primary Health-Care Centre, Institut Català de la Salut, 43005 Tarragona, Spain; 3Pharmacy Unit, Gerència d’Atenció Primària Camp de Tarragona, Institut Català de la Salut, 43005 Tarragona, Spain; 4Research Support Unit Tarragona-Reus, Institut Universitari D’investigació en L’atenció Primària Jordi Gol, (IDIAP Jordi Gol), Institut Català de la Salut, 43202 Reus, Spain; esatue.tgn.ics@gencat.cat (E.M.S.-G.); crey.tgn.ics@gencat.cat (C.R.-R.); fmartin.tgn.ics@gencat.cat (F.M.M.-L.); 5Faculty of Medicine and Health Sciences, Universitat Rovira i Virgili, 43201 Reus, Spain

**Keywords:** primary care, patient safety, incident notification, risk management

## Abstract

Objectives: (1) To describe the epidemiology of patient safety (PS) incidents registered in an electronic notification system in primary care (PC) health centres; (2) to define a risk map; and (3) to identify the critical areas where intervention is needed. Design: Descriptive analytical study of incidents reported from 1 January to 31 December 2018, on the TPSC Cloud™ platform (The Patient Safety Company) accessible from the corporate website (Intranet) of the regional public health service. Setting: 24 Catalan Institute of Health PC health centres of the Tarragona region (Spain). Participants: Professionals from the PC health centres and a Patient Safety Functional Unit. Measurements: Data obtained from records voluntarily submitted to an electronic, standardised and anonymised form. Data recorded: healthcare unit, notifier, type of incident, risk matrix, causal and contributing factors, preventability, level of resolution and improvement actions. Results: A total of 1544 reports were reviewed and 1129 PS incidents were analysed: 25.0% of incidents did not reach the patient; 66.5% reached the patient without causing harm, and 8.5% caused adverse events. Nurses provided half of the reports (48.5%), while doctors reported more adverse events (70.8%; *p* < 0.01). Of the 96 adverse events, 46.9% only required observation, 34.4% caused temporary damage that required treatment, 13.5% required (or prolonged) hospitalization, and 5.2% caused severe permanent damage and/or a situation close to death. Notably, 99.2% were considered preventable. The main critical areas were: communication (27.8%), clinical-administrative management (25.1%), care delivery (23.5%) and medicines (18.4%); few incidents were related to diagnosis (3.6%). Conclusions: PS incident notification applications are adequate for reporting incidents and adverse events associated with healthcare. Approximately 75% and 10% of incidents reach the patient and cause some damage, respectively, and most cases are considered preventable. Adequate and strengthened risk management of critical areas is required to improve PS.

## 1. Introduction

Patient safety (PS) is the quality of care dimension that aims to reduce and prevent risks associated with healthcare [1]. In 2004, the World Health Organisation (WHO) created the program The World Alliance for Patient Safety to develop policies to improve patient care in health services [2]. Currently, PS is considered a key quality factor of healthcare, and the detection and alleviation of incidents derived from the delivery of care have become a priority for most health systems [3]. In Spain, the Ministry of Health, Social Services and Equality, in coordination with the regional governments, launched the national PS strategy to promote a culture of PS among professionals and patients and to develop and encourage research on PS [4]. Consequently, most autonomous communities currently have electronic systems to report PS incidents, although the implementation in the primary care services remains uneven [5].

Regarding PS, the ideal target would be zero damage. However, zero damage is not a feasible goal because, while no patient should suffer any harm derived from the healthcare received, in clinical practice some events are considered unpreventable [6]. Instead, organisations should undertake the responsibility to learn from their own errors, and the first step toward reducing the incidence is the notification of all detected PS incidents [7,8]. The 2000 report of the IOM-Institute of Medicine (To err is human) called, encouraged “*healthcare organisations to participate in voluntary reporting systems as an important component of their patient safety programs*” [9]. Good practice toward improving PS culture within an organisation includes encouraging the notification of incidents in an environment of trust [10]; analysing the incidents (types, frequencies, causes and effects in patients) is essential to improve error detection and to develop preventative interventions [11].

Since the publication of the IOM-Institute of Medicine report, a growing international interest has underscored the need for accurate information on the frequency, severity and characteristics of PS incidents, mainly in high-risk situations [12]. Voluntary incident reporting systems constitute a crucial source of information, and although no standardised, validated methods to detect the most important incidents have been yet defined, a combination of proactive and reactive approaches is generally recommended [13]. A recent meta-analysis indicated a pooled prevalence of PS incidents (preventable and non-preventable) of 12% [14]. This review stressed the lack of primary care studies on PS (only 4% of the studies analysed in the article were conducted in the primary care setting). This might respond to the false perception that less adverse events occur in primary care compared to more specialised levels of care, which only delays the culture of safety [15]. As a result, even if most patients are regularly visited in primary care, PS research has mainly focused on hospitals [16]. To further stress the need to promote research on PS in primary care, it is widely recognised that relevant aspects of care such as patient characteristics, organisational structure and relationships between professionals and patients differ between primary care and hospital, and therefore it cannot be assumed that the patient risks and reported results of PS incidents will be the same in both settings [17].

In 2012, the WHO constituted the *Safer Primary Care* project, a taskforce to study issues related to PS in primary care [16]. In 2016, a WHO-led systematic review to determine the frequency, severity, and preventability of PS incidents in primary care, noted the lack of robust large-scale studies available [18]. This review highlighted the APEAS study, which analysed over 96,000 primary care visits in Spain and concluded that approximately 1 in 100 visits was associated with a preventable adverse event [19]. In this study, although medication (adverse drug reactions and medication errors) was the main source of error related to PS, many incident notifications referred to miscommunication, management and other organisational factors. Since the results of this study were published nearly a decade ago [20], clinical practice has become more complex and has incorporated more technological advances. Combined with a large number of contacts with patients, it would be reasonable to assume that the epidemiology of PS incidents has also changed in our setting. In fact, a recent systematic review recommends the periodic evaluation of PS incidents in primary care, to evaluate the efficacy of safety measures in health institutions [21].

The general objective of this study is to obtain up-to-date data on the epidemiology of PS incidents retrieving the types, characteristics, contributing factors and consequences from the records of an incident notification system implemented in a Primary Care Health Region of Catalonia, Spain. The results should define the current PS risk map, identifying critical areas susceptible to corrective interventions to reduce the incidence of adverse events and to improve PS.

## 2. Materials and Methods

### 2.1. Study Design

Descriptive cross-sectional study with analytical components, based on the electronic records of a system (TPSC Cloud™ platform V8.0) for the notification of PS incidents that occurred from 1 January to 31 December 2018, in the primary care setting of the Tarragona health region (Spain). This study corresponds to the first phase of the mAPaSP project (Effectiveness of the implementation of different patient safety tools to define a new risk map in PC) to improve the quality of care. The project is funded by the Research Fund of the National Health Service (code PI17/02063).

### 2.2. Setting

The Tarragona health region of the Catalan Institute of Health (ICS in its Catalan acronym) serves a population of about 320,000 inhabitants and employs approximately 1245 professionals. It includes 20 primary care centres, two sexual and reproductive healthcare centres (ASSIR in its Catalan acronym) and two emergency care centres (CUAP, in its Catalan acronym).

The Spanish Primary Care system differs from that of other countries in the European Community and the United States. The Spanish National Health System provides universal health coverage, is divided into 18 autonomous health systems and is organised in health regions. In the Catalan Health Service, each health region has several primary care centres and at least one referral hospital. The health centres are managed with public funds and owned by the regional health authority. Each primary care centre has a multi-disciplinary team that consists of family doctors, paediatricians, nurses, dentists, physiotherapists, psychologists, technical assistants and administrative staff. The centres provide general care in the community to patients of all ages, who attend appointments and/or urgent consultations. Health education and promotion, elderly care, adult care, mental healthcare, children’s health, women’s health and reproductive health are included in the services offered by primary care centres. In general, there is a high provision of these services [22]. All visits are recorded with the use of electronic medical records integrated into a clinical workstation (eCAP), which also supports medication prescription, which allows certain medication errors to be detected online [23].

### 2.3. Reporting System

In Catalonia, the Health Plan of the Health Department includes PS (strategic line 8, *Management of excellence and safety*), a strategy that has been incorporated into most health centres [24]. In accordance with the Accreditation Model and international PS guidelines, all organisations of the public health network attached to the Department of Health of the Generalitat de Catalunya have a notification system of PS incidents for quality improvement and clinical safety [25,26]. To this end, primary care centres and hospitals use the Platform for Patient Safety Management with the TPSC Cloud™ software, a centralised reporting system accessible from any health centre, which facilitates the notification and management of all types of PS incidents [27]. The main features of the TPSC Cloud™ application are as follows [28]:-Notification: a structured record is generated. There are different forms, which correspond to the type of incident reported.-Management: encourages proactive incident analysis to detect flaws and improve processes. Tools such as the risk matrix, cause-effect, process and root cause analyses are available for professionals responsible for incident management [29]. For instance, the risk matrix evaluates the risk of an incident based on the probability of occurring and the impact on the patient. These two criteria are represented in a table where each box colour is related to risk severity (green colour indicates low risk, yellow moderate risk, and red high risk). For the cause-effect analysis, we design a fishbone diagram (Ishikawa diagram) describing the problem and the main underlying causes. The root-cause-analysis is a systematic process to determine the underlying factors that have contributed to the occurrence of the incident, particularly the analysis of latent conditions (systems and processes). The end goal is to sensibly respond to the following questions: What happened?; Why did it happen?; Can we prevent it from happening again?-Analysis and reports: the platform analyses incidents to systematically identify risks and prevent errors.-Improvement actions: allows planning and monitoring of improvement actions, preventative measures or changes in the organisation.

### 2.4. Incident Notification Procedure of General Practices and Data Collectors

The Tarragona Health Region of the Catalan Institute of Health dedicates specific resources to reporting and analysing PS incidents affecting their health centres. It also offers an e-learning program that teaches all health professionals how to report PS incidents. The PS application is accessible from the corporate website intranet, where any professional can submit a notification after a secure login. Although professionals do not need to report incidents in the notification system, they are advised to record any possible PS incident, regardless of whether the patient has suffered any type of harm. The reports are confidential, non-punitive, and are entered voluntarily. Each report includes patient and healthcare professional data to facilitate follow-up. Each case is assigned an individual identification code for analysis. However, once the incident is closed, the information is stored anonymously.

All health centres enter the information on PS incidents using a structured form accessible from the TPSC Cloud™ platform on the corporate intranet of the Tarragona Regional Management of the Catalan Institute of Health (http://camptarragona.cpd2.grupics.intranet/ accessed on 1 July 2021). This information is automatically stored (including a backup copy). Access to the management system is password protected.

Following recommendations by the WHO World Alliance for Patient Safety [29], the notification system guides the notifier through a series of screens with drop-down responses designed to collect standardised incident information. In Appendix A, we collect and describe the variables in the notification registry of PS incidents.

The report consists of six structured fields to record the healthcare unit and professional group of the informant, the type of safety incident based on the WHO classification (for example, administration, medication), consequences for the patient and the organisation, severity of the incident, causal factors (related to communication, organisational planning, or healthcare delivery), contributing factors (for instance, patient-dependent such as comorbidity, professional-dependent such as workload, or physical environment), improvement actions and corrective measures are undertaken to reduce the risk. Incidents are simultaneously classified according to the international classification of the WHO (10 categories) [30], and the Accreditation Model of the Department of Health of the Generalitat de Catalunya (15 categories) [31]. Most information fields in the form are mandatory, and some allow multiple entries (such as causes or contributing factors) to obtain detailed information. The form adds four free text fields, where the informant can describe the incident, contributing and/or mitigating factors and improvement suggestions. The data entry process takes an average of 10 min to complete. The application allows immediate access to the report after data entry, to amend the information and to verify and follow-up the PS incident. To ensure correct completion of the incident notification form, professionals have access to an electronic PS learning program accredited by the Department of Health of the Generalitat de Catalunya. In addition to theoretical concepts (definition of an incident, adverse event, types of incidents, contributing factors, etc.), the program discusses several examples to illustrate how the notification system works. A user manual is also provided.

### 2.5. Taxonomy and Definitions

A “patient safety incident” was defined as any event (or circumstance) during an episode of patient care that had the potential to (near miss) or actually caused injury or harm (adverse events) to the patient. According to the WHO proposal, it also includes the evaluation of the severity of the damage and the preventability of the incident [30].

The severity of the damage is firstly classified by the notifier according to a risk matrix (very low, low, moderate, high and extreme), and then reclassified using the definitions of the International Classification for PS in Primary Care [7]:

(1) no harm caused by the reportable safety circumstance or incident with the potential to cause harm, which was prevented by a timely intervention (near-miss); (2) no harm caused, although the incident was not prevented; (3) adverse event with mild damage, with minimal symptoms which required additional observation or minimal treatment or intervention; (4) adverse event with moderate damage, the incident caused symptoms, treatment was required and caused significant but not permanent damage (for instance, hospitalization); (5) adverse event with severe damage, intensive treatment or major intervention was required, or resulted in permanent damage such as disability, or lasting physical or mental sequelae, and (6) extreme adverse event, for a situation that caused death or near-death.

The preventability of the PS incident was classified by the notifier as: potentially preventable, not preventable or unclear.

### 2.6. Notification Evaluation Procedure

Each PS notification follows a comprehensive evaluation process in agreement with national and international guidelines [30]. The Primary Care Health Area of the Catalan Institute of Health of Tarragona established Patient Safety Functional Units in all its healthcare centres in 2011 to analyse reported PS incidents, to identify critical areas and to provide improvement actions [32]. Each Unit has a local clinical safety leader, the professional responsible for reviewing and classifying PS incidents. Duplicate reports are removed from the database. Staff in the PS Functional Unit of Primary Care centres can consult with the experts in the Central PS Unit.

The Central Safety Functional Unit of the Primary Care Management consists of eight professionals experts who have received academic training in PS (postgraduate or master) and with extensive experience in PS incidents management. In addition, each of them contributes specific knowledge in certain fields (clinicians, pharmacologists, nurses, administrative officers and statisticians). All reported incidents are regularly and systematically reviewed by two or more of these experts, who check consistency and compliance with the classification principles and assess preventability using a peer review system (the agreement between the two experts is determined). For the assignment of the reviewer number and profile, the type of incident is taken into account (for example, medical or nursing care, administration, medication, etc.). In case of disagreement, the two reviewers try to reach a consensus and if this fails, another expert intervenes until an agreement is reached. The Central Safety Functional Unit also regularly analyses the incident notification system and provides reports to the safety officer in the health centres on the quality of their notifications, to strengthen the safety culture of the organisation. It also provides methodological support to health centres during analyses (root cause analysis) and defines improvement actions and safe practices to be implemented in all centres. In addition, it periodically publishes health alerts and bulletins with the identified risk areas.

### 2.7. Statistical Analysis

The statistical package SPSS version 19.0 (IBM Corp., Armonk, NY, USA) was used for analysis. No data referring to patients or professionals were included, only the primary care centres were identified to perform some aggregate analyses.

We conducted a descriptive exploratory analysis using absolute and relative frequency distributions in crosstabulations, to define the proportion of total and specific PS incident types according to severity (incidents that do not reach the patient, incidents without harm and events that cause harm). We carried out a specific analysis of adverse events and assessed possible differences (with respect to the global number of reported incidents) in the distribution of variables and dimensions included in the TPSC Cloud™ questionnaire. Differences in proportions were compared using the chi-square test. Statistical significance was set at *p* < 0.05 (two-tailed).

We analysed causal factors associated with preventable PS incidents to identify risk areas (risk map). The advice on critical areas derived from the analysis of contributing factors was based on the prioritization carried out by The Central Safety Functional Unit of the Primary Care Management experts according to the literature [30].

## 3. Results

During the study period (1 January to 31 December 2018), the TPSC Cloud ™ incident notification system of the Tarragona Primary Care Region of the Catalan Institute of Health contained a total of 1544 records. After review by the Patient Safety Central Functional Unit, 68 notifications (4.4%) that failed to describe true PS incidents were excluded. 347 notifications (22.5%), that according to the Unit’s experts’ criteria, did not have sufficient information for correct coding or because the adjudication of the evaluation was pending analysis. Finally, the analysis included 1129 (73.1%) PS incidents considered resolved. Figure 1 shows the flowchart of the study and the distribution of the type of incidents reported according to severity (WHO classification).

The notifications varied widely by the centre (from 7–135 cases, 0.6–12.0%). When standardizing for the population served by the centre, the range of incidents was also wide, with a median of 3.7 per 1000 patients visited (1.2–18.7‰). Globally, the notification rate in all primary care centres was 2.38 incidents per 1000 visits. Significantly more PS incidents (*p* < 0.01) were reported by the 20 primary care centres (77.5%; 95% CI 75.0–79.8), compared with the two ASSIR centres (11.7%; 95% CI 10.0–13.7) and the two emergency care centres (10.8%; 95% CI 9.1–12.8). These differences did not change when comparing notification rates between primary care centres (3.25‰ for a total of 269,534 visits) with the ASSIR centres (1.13‰ for a total of 116,316 visits) and the emergency care centres (1.37‰ for a total of 89,286 visits).

Comparing occupations, nurses and nursing assistants reported significantly more incidents (48.5%; 95% CI 45.5–51.4) than doctors (32.6%; 95% CI 29.2–35.4), admission and management staff (15.9%; 95% CI 13.9–18.2) and other services (3.0%; 95% CI 2.2–4.2), (*p* < 0.01). Regarding adverse events, they were significantly reported mostly by doctors (66.7%; 95% CI 56.8–75.3) compared to nurses and nursing assistants (20.8%; 95% CI 13.9–30.0) and admission and management personnel (6.5%; 95% CI 2.9–13.0), (*p* < 0.01).

Out of the 1129 records analysed, 75.0% were incidents that reached the patient (with or without damage), and the remaining 25% did not reach the patient (notifiable and near-miss circumstances). Most (91.5%) were precursor safety incidents (did not reach the patient or there was no detectable damage). Adverse events represented 8.5%, incidents leading to treatment or hospitalization and causing temporary injury reached 4.1%, and in 0.5% of cases severe incidents reached the patient and caused severe damage or a situation close to death. Table 1 shows the distribution of reported incidents according to severity and repercussions for the patient.

Table 2 shows the distribution of the type of incidents notified according to risk (risk matrix). Over half of incidents (64.9%; 95% CI 62.0–67.6) were low or very low risk cases (only in exceptional circumstances and with a very low frequency), and 3.5% (3.5%; 95% CI 2.5–4.7) were considered high or extreme risk (require detailed analysis and immediate corrective measures).

### 3.1. Report Classification

Figure 2 shows the distribution of PS incidents categorised according to the WHO international classification (Figure 2a,b).

Overall, the most frequent incidents related to clinical-administrative management (31.7%; 95% CI 29.1–34.5) and clinical management-procedures (25.8%; 95% CI 23.3–28.4), followed by medication-related incidents (18.7%; 95% CI 16.5–21.1). In contrast, incidents associated with analogue and digital documentation, infrastructures and facilities, healthcare equipment and devices, patient behaviour, falls and other accidents, and infection associated with healthcare, were relatively uncommon (*p* < 0.001). Similarly, the most frequently reported adverse events were also related to clinical-administrative management (33.3%; 95% CI 24.7–43.2) and clinical management-procedures (37.5%; 95% CI 28.5–47.5) and medication (18.8%; 95% CI 12.2–27.7).

Table 3 shows the distribution of PS incidents categorised according to the classification of the Department of Health of the Generalitat de Catalunya. Here, the most commonly overall reported PS incidents were related to administrative procedures (22.9%; 95% CI 20.6–25.5), laboratory (21.6%; 95% CI 19.3–24.1) and medication use (16.8%; 95% CI 14.3–18.9). Importantly, most of these incidents did not reach the patient or did not cause harm. However, the analysis of adverse events revealed that the most frequent were related to medication use (18.8%; 95% CI 12.2–27.7), diagnostic imaging (16.7%, 95% CI 10.5–25.4) and continuity of care (13.5%; 95% CI 8.1–21.8).

### 3.2. Causal and Contributing Factors

Figure 3 presents the distribution of causal factors as a risk map based on the five critical classification areas proposed in the APEAS study, for all the incidents recorded (Section A), and for the adverse events (Section B).

Communication was the area that concentrated the highest number of PS incidents (27.8%; 95% CI 25.3–30.5), followed by management (25.1%; 95% CI 22.6–27.7), healthcare (23.5%; 95% CI 21.1–26.0) and medication (18.4%; 95% CI 16.3–20.8); diagnosis had a reduced representation (3.6%; 95% CI 2.7–4.9). The risk map for adverse events showed a high representation of care delivery (33.3%; 95% CI 24.7–43.2), followed by communication (20.8%; 95% CI 13.9–30.0), management (17.7%; 95% CI 11.4–26.5) and medication (15.6%; 95% CI 9.7–24.3); the diagnosis was again the less represented area (11.5%; 95% CI 6.5–19.4).

In all reported incidents, the most prevalent contributing factors were related to the professional (55.8; 95% CI 52.9–58.7), the organisation (36.8; 95% CI 34.0–39.6) and external (25.9; 95% CI 23.5–28.6). Similarly, for adverse events the main contributing factors were: professional (47.9; 95% CI 38.2–57.8), organisation (40.6; 95% CI 31.4–50.6), and external (33.3; 95% CI 24.7–43.2). In both cases, factors related to the environment and the patient had a low prevalence (Appendix A).

Figure 4 shows the analysis of contributing factors for all PS incidents according to the WHO classification. PS incidents are categorised into those that did not reach the patient, PS incidents without damage, and adverse events.

With regard to preventability, only 9 PS incidents were classified as non-preventable (0.8%; 95% CI 0.42–1.5).

### 3.3. Resolution Level and Proposed Actions

Table 4 shows the information on the resolution level of PS incidents. The suggested solution to the PS incident was similarly distributed between the Patient Safety Functional Unit of the healthcare centre and external units (46.8% [95% CI 43.9–49.7] and 53.2% [95% CI 50.3–56.1], respectively).

The TPSC Cloud™ notification system contains relevant information on methods to prevent incidents from happening again. Up to 35.8% of PS incidents (95% CI 33.0–38.6) did not generate any action. The most common suggestions were to inform the staff of the centre where the incident had occurred, conduct some training or review procedures. Only 3.3% of incidents (95% CI 33.0–38.6 2.4–4.5) warranted in-depth analysis and a more specific improvement intervention. In all these cases, the measures were mainly related to the most severe incidents. Table 5 describes the frequency of improvement measures and actions to reduce risk.

The main proposals were:Creation of multidisciplinary improvement teams to standardise clinical practice. For example, the adequacy of diagnostic tests or pharmacological treatments (such as oral anticoagulant therapy), the implementation of checklists that allow a structured and daily briefing to be carried out in the health centre and each reference laboratory, or the standardization of various administrative processes.Continuing professional development yearly. 95% of the professionals in primary care centres have received PS training. This training takes place yearly and consists of a 6-h workshop accredited by the Institut Català de la Salut. Furthermore, all professionals of the Central Patient Safety Functional Unit have been trained in the management of adverse events and in relation to second and third victims. These concepts acknowledge that “harm from PS incidents does not always stop with patients and their families” (considered the first victims of the error), since often it is “the healthcare workers involved in an incident, who can also experience significant harm” (second victims) and, even “those with indirect exposure to an adverse event can become victims (third) of an adverse event” [33]. A webpage is available in Spain with information regarding second victims (http://www.segundasvictimas.es/index.php, accessed on 12 August 2021), and the Department of Health of the Generalitat de Catalunya provides online training accessible by all health organisations (http://seguretatdelspacients.gencat.cat/ca/professionals/formacio/gestio_de_riscos/segones-victimes/, accessed on 12 August 2021).Creation of transversal PS units to analyse incidents related to communication between different healthcare services.Root-cause analysis of severe adverse events related to diagnostic delay in cancer patients, lack of coordination between different levels of care and control of narcotics in health centres.Publication of patient safety bulletins with general and specific content on adverse events, in relation to communication, laboratory and safe use of medication. In reports on health warnings, such as the case of necrotizing fasciitis due to simultaneous intramuscular administration of metamizole and diclofenac.

## 4. Discussion

### 4.1. Summary of Main Findings

Research in PS has increased within healthcare organisations with the raising awareness on medical errors and adverse events [14]. While previous studies on medical errors have been conducted in Spain [19,20], the current study provides up-to-date data on the types, causes and contributing factors of the voluntarily reported incidents related to primary care activity, to define a new risk map for PS.

The first step to prevent PS incidents is to obtain evidence on the frequency, type, impact, and causal or contributing factors [1]. The implementation of a voluntary incident reporting system is considered fundamental for recording and processing information related to PS [34]. In this study, we describe the incidents voluntarily reported using an electronic registry (TPSC Cloud™) in 24 primary care centres in the Tarragona region of Catalonia (Spain). The most common notifications relate to incidents that reached the patient (75%), although only a small percentage caused adverse events. The adverse events reported were usually mild, and only a small fraction (0.5%) resulted in permanent damage or were life-threatening. A minority of incidents (3.5%) were categorised as high or extreme risk, and most (99.2%) were considered preventable. Our data confirm that despite the wide variety of contributing factors, those related to the healthcare process represent the main sources of PS incidents. Based on the relative contribution of incident types, we present a visual model (risk map) to determine the critical areas to prioritise to reduce damage, in this case, care delivery (33.3%), communication (20.8%) and clinical-administrative management (17.7%).

### 4.2. Comparison with Existing Literature

PS incidents are uncommon in primary care (2–3% of visits) compared to 10% of hospitalizations [18]. In our study, the global notification rate was 2.4 incidents per 1000 visits, although the analysis by health centres revealed that the rate was larger in primary care centres when compared with the ASSIR and the emergency care centres. However, these data become relevant when taking into account the high number of consultations carried out daily [35,36]. Notably, reporting rates vary substantially depending on the care facility. In the aforementioned APEAS study, 73.5% of adverse events occurred in a primary care centre, compared with 25.8% of hospital care [19]. In our study, the incidents originated mostly from primary care centres (77.5%), but we observed a higher incidence of adverse events in the primary care emergency services, which doubled the incidence of other primary healthcare units and generated 20% of all adverse events. Other studies have already documented a greater number of adverse events in emergency services, which might be explained by the higher complexity of patient presentations [37].

The culture of safety depends on all primary care professionals. In this study, the notification system was accessible to all health professionals, but nurses and nursing assistants contributed to almost half of the total notifications, compared to doctors and admin staff. It has been previously reported that in primary care centres, nurses (*mainly women in their fifties*) have a higher awareness of PS [38]. In a study conducted in Catalonia between May 2013 and December 2016, nurses were responsible for the notification of 62% of PS incidents [27]. Interestingly, variation in reporting rates between nurses and doctors can be attributed to different perceptions of what constitutes a PS incident or adverse event and crucially, to different attitudes towards reporting [8]. This might explain why most adverse events were reported by doctors. In any case, we should underscore that a higher notification rate may not correlate with worse patient care, but rather with an institutional culture that encourages error notification without fear of negative consequences for the professionals [39]. In our setting, all healthcare professionals receive training and are encouraged to report PS incidents. However, doctors might still work under a code of beliefs of self-blame and personal responsibility and thus with concern for personal repercussions, rather than interpret the adverse events and complications derived from healthcare as the final event in a series of organisational errors.

International comparisons of PS incidents can be difficult because of the variation in methods and classifications used in different primary care systems [18]. Indeed, a recent WHO review identified 21 different methods for classifying the severity of PS incidents [7]. In our study, we used the WHO classification and included not only adverse events, but all reported PS incidents, whether they reached the patient or not, as we believe that the analysis of reportable circumstances and near misses providing relevant information to identify potential sources of harm [40]. Most PS incidents assessed in this study did not reach the patient and when they did, they did not have a negative impact (91.5%), as observed in other primary care studies [18]. Notably, the proportion of adverse events was relatively low (it represented only 8.5% of all incidents voluntarily reported), compared to the higher number reported in hospitals [14]. This can be explained by the low number of severe errors in primary care, and because the number of people using primary care services is much higher than people using hospital services [1]. When thinking that this proportion is high, we should consider the bias caused by underreporting incidents that do not reach the patient or do not cause harm.

The majority of PS incidents that occur in primary care do not pose severe risks to patients [20]. However, assessing the severity is complex, because incidents are not uniformly classified in studies conducted in primary care [7]. In a systematic review, Panesar et al. concluded that approximately 4% of incidents cause severe harm, defined as having a significant impact on the well-being of the patient, but the range was very wide (<1% to 44%) [18]. In our study, half (47%) of all adverse events did not cause injury or only required observation, a third (34.4%) required additional treatment which was indicated in the same health centre, 13.5% of cases required hospitalization, 3.1% cases suffered a permanent injury and 2.1% a near-death event. In contrast, O’Beirne et al. reported that only 9% of primary care incidents were moderate or severe, and 1% of permanent severe impact [41]. In a recently published study in primary care in Spain, Guerra-Garcia et al. concluded that almost 50% of adverse events produced mild effects and only 2.4% were critical [42]. Our results differ from a systematic review by Panagioti et al. which observed a pooled proportion of events classified as severe or near-death of 12% (9–15%) [14]. To understand the differences reported, it should be noted that many PS incidents in primary care go unnoticed because they have no consequences and are thus not reported. Some studies overestimate the severity of these events because professionals may only notify incidents that reach the patient and cause some type of harm [18].

It is accepted that PS incidents and adverse events are usually replicated in similar institutions with similar causes and contributing factors, depending on the level of analysis [43]. The most common causes in primary care have been previously indicated and can be classified as incidents due to miscommunication between professionals and the patient, errors in administrative management, care delivery, diagnostic processes and prescription and medication management [44,45]. Although the most frequent causes vary depending on study characteristics (retrospective or prospective), data collection methods, definitions used and population studied, the most problematic areas are administration and communication, errors in diagnosis and the prescription and management of medications. In our study, the descriptive analysis illustrated the importance of communication, administrative management and care delivery regarding the total number of notifications and adverse events, representing more than 75 and 70% (respectively) of the causal factors involved. These results clearly differ from the APEAS study, in which medications and diagnosis represented >60% of the most frequent causal factors of adverse events in primary care [19,20]. A more recent study by Guerra et al. concluded that most adverse events that reached the patient were related to medication, diagnostic tests and laboratory errors [42].

In primary care, miscommunication is one of the main causes of error [46]. Miscommunication was also very relevant in our study regarding incidents and adverse events (27.8% and 20.8%, respectively). Consequently, communication has been prioritised in primary care PS research strategies [47,48]. Administrative management was the second critical area, with 25.1% of reported incidents, mainly due to appointment errors and long waiting lists in the case of adverse events. In relation to adverse events, delivery of care was the most common PS cause (33.3%), particularly inadequate management of the procedure and the patient.

Medication-related PS incidents are reported as common and often preventable errors in primary care [20]. Medication-related errors are the fourth most frequent critical area in our study, representing 18% of reported incidents and 15% of adverse events, and are mainly related to ineffective prescription, wrong medication, wrong dosage and errors in manipulation and preparation. A recent systematic review observed that incidents related to medication and other treatments represented the highest proportion of preventable harm to the patient [14]. Electronic prescriptions significantly improve medication safety in primary care [49]. However, we cannot solely rely on electronic prescription, as it alone is not enough to prevent all prescribing errors [50,51].

The preventability of PS incidents described in the scientific literature varies and includes hospitalised patients, patients treated in emergency services and primary care centres. Panagioti et al., in a systematic review that included pooled data from 70 studies published since 2000 in a wide range of medical settings globally, concluded that half of the patient incidents are preventable [14]. Although most evidence on preventable patient harm originates from hospital data, it is believed that most PS incidents that occur in primary care are preventable [52]. Corroborating these results, we determined that most (99.2%) reported incidents could have been prevented (as previously shown in the section results).

The literature review conducted by Makeham et al. for the WHO concluded that 70–76% of instances of patient harm could be prevented [53]. In Spain, the results are variable, ranging from 70.2% of clearly preventable incidents reported in the APEAS study to 94.6% described in a more recent study carried out in a primary care service [19,42]. In these studies, preventability seems to correlate with severity, i.e., the higher the severity, the more preventable the incidents would be. The concept of preventability has probably evolved over time and is related to a cultural change in professionals: the fact that they are preventable is seen as an opportunity for improvement and greater awareness of safety as a new dimension of quality of care.

Although there is limited evidence on the effectiveness of strategies to improve patient safety, including incident reporting systems, the general advice is to combine proactive and reactive methods [13,54]. A recent study conducted in a tertiary hospital in Spain concluded that the improvements based on incident-reporting system data and analysis of PS incidents by workshop-trained local clinical safety leaders resulted in a significant reduction in PS incidents, particularly those related to medication categories, communication and technology [55]. However, data and analysis of PS notifications are meaningless unless they drive improvement. The information contained in PS notification forms is the starting point for the clinical risk management units to introduce continuing improvements in health care. [56]. The implementation of a reporting system guarantees a virtuous cycle of learning, training and reallocation of resources [57]. In agreement with the recommendations of the WHO expert group, we consider our new PS incident notification system as the starting point of an improvement process to prevent PS incidents and associated damages. The combination of the tools offered by the platform together with the manager’s expertise should result in the implementation of preventative measures at the individual, service and organisational level [7]. A recent Cochrane systematic review concluded that this combination of actions is effective in reducing preventable PS incidents [58].

### 4.3. Major Strengths and Limitations of This Study

This is a large-scale study to investigate PS incidents in primary care. To carry out this, we conducted a 12-month observational study that included a comprehensive review of a significant number of voluntary notifications from primary care professionals. To the best of our knowledge, no study has described to date the frequency and types of primary care PS incidents and adverse events voluntarily reported in our setting.

The reporting system used, the TPSC Cloud™, is integrated as a voluntary incident reporting system tool for PS management and organisational learning in all of our primary care centres. Voluntary incident reporting systems are the most common data source in PS studies due to speedy data availability at a reasonable cost [53]. They can also provide accurate descriptions of PS incidents and underlying contributing factors which might assist the redesign of the clinical process [59]. Although some authors worry that the data contained in these records is reactive and thus does not represent all incidents that occur during patient care [60], the real-life information they provide is clearly significant [14]. We should emphasise that healthcare professionals have highly rated these reporting tools in terms of usability and reliability [61].

To enhance representativity, this study has a pragmatic design implemented with methodological rigour while facilitating the inclusion of notifications from various primary care health services: regular consultations; emergency services; health promotion; well-child care; and women’s health. Furthermore, no additional staff was recruited to increase reporting to minimise the Hawthorne observation bias due to participation, and also to demonstrate the feasibility of routine data collection on PS incidents in primary care daily practice [18,20]. This is relevant regarding PS challenges encountered in community medicine, which might require different improvement strategies [35].

The sample size was comparable to similar studies. In a recent systematic review on PS in a wide range of global medical settings, sample sizes presented a median of 1440 patients (range 128–96,047) [14]. Although errors in voluntary reporting systems might impact data quality, we only excluded 4.4% of reports because they were not PS-related incidents. This suggests that most participants understood the definition of PS, which is undoubtedly linked to the access to training in PS provided in our setting. Training to adequately report incidents is considered crucial to improve PS in primary care [62,63]. PS incident reporting requires training and skills since the effective classification of notifications depends on the judgment of the person who reports the incident, which can vary depending on the occupation, knowledge of the system and past experience [7]. Future research should minimise a possible training bias [64].

Overall, PS cannot be considered (more or less) a universal phenomenon in primary care, but rather a cluster of specific country research phenomena. Similarly, there is no universally accepted standard classification system for PS incidents in primary care, a widely reported limitation by PS researchers [7]. However, the use of international standards is crucial to enable the comparison of results and research contexts. In this case, we have used the WHO International Classification of PS. We have also included the classification of causal factors suggested in the APEAS study, which is a reference within Spain. While we use the WHO classification to internationally standardise our data, we find the APEAS classification better suited for clinical practice.

This study presents several methodological limitations that warrant caution in the interpretation of results. Firstly, the selection bias is associated with the voluntary nature of the notification. Despite the accessibility of the TPSC Cloud™ electronic record system to all primary care professionals, we believe that some PS incidents, particularly the mild that do not cause harm or injury, are not reported and might be even normalised when they occur repeatedly [7]. Nonetheless, notification systems remain a crucial instrument for quality systems, since they point at priority areas of intervention to formulate guidelines for the improvement of PS in clinical practice [11]. Although we cannot rule out underreporting, we consider that our sample size, which includes a wide range of types and severities of PS incidents, is representative. Other studies have reported a high degree of compliance with notifications using voluntary systems [7,40]. Future studies should take this limitation into account, and incorporate parallel methods for assessing PS risks in primary care [64].

The quality of the records, i.e., PS reports with incomplete or missing data, is another limitation of the study, since it affects the analysis of notifications and the representativeness of the recorded incidents. In this study, 23.5% of incidents were excluded because the information was insufficient for correct coding or because the final result of the evaluation was pending confirmation. Some previous studies have reported that almost a third of the notifications were rejected because they were not PS incidents or were insufficiently defined [44]. For situations with insufficient information, relatively frequent in clinical practice, the new Primary Care Harm Severity Classification System of the WHO proposes a specific category for incidents in which “an error occurred, but no damage could be determined” [7].

In this study, PS notification was voluntary and usually submitted during regular working hours, which might have caused underreporting or less detailed reporting of the data, particularly taking into account that primary care professionals have practically no time for non-clinical activities. Other factors that might contribute to underreporting are fear of sanctions and legal consequences, concern about the anonymity and confidentiality of the information, and uncertainty about the relevance of the notification. We should underscore that the ICS-Catalan Institute of Health promotes a culture for the improvement of PS by which staff are encouraged to report PS incidents and professionals are expected to speak frankly [25]. Lastly, we are aware that the generalization of our findings might be hindered by the conceptual variability of PS in primary care and the quality of the records.

Finally, our study does not analyse the factors associated with healthcare professionals, a significant limitation since certain attitudes and behaviours of professionals might be associated with PS incidents in primary care [44]. Additionally, some characteristics of the patient, such as older age and a high number of comorbidities have been associated with a greater risk of adverse events [65].

### 4.4. Implications for Clinical Practice and Future Research

To the best of our knowledge, this is the first study on primary care PS notifications in Catalonia. Although we strongly suspect underreporting (only the tip of the iceberg appears) [64], this analysis of notifications focused on causal and contributing factors opens new opportunities to learn from PS incidents and improve clinical safety in primary care and underscores priority areas for action: communication failures between patients and professionals, errors in management, care delivery and medication management.

This study should promote the advance of PS in our health services. The results underscore the importance of reliable PS assessments and of expanding the culture of safety in healthcare facilities. It is also important to take into account that while reactive incident reporting provides crucial information, it should be complemented with proactive methods that assess the views of patients and health professionals [60]. Future studies should assist primary care managers in planning and developing organisational strategies aimed at improving the quality of care provided.

## 5. Conclusions

PS incident-reporting software, such as the TPSC Cloud™ platform, are considered adequate systems for recording incidents and adverse events in healthcare facilities. Awareness type and severity of incidents, their causes and contributing factors, and critical areas are crucial for the effective management of PS. The information generated by these tools is the starting point for the activities of the PS Functional Unit and quality and safety leaders, which should translate into specific measures to reduce adverse events and improve clinical care.

## Figures and Tables

**Figure 1 ijerph-18-08941-f001:**
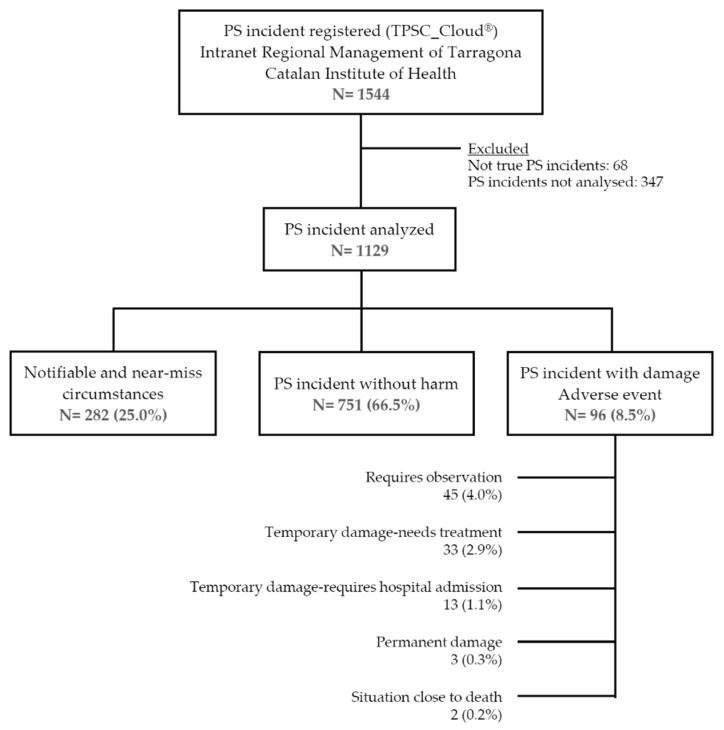
Flowchart illustrating how patient safety (PS) reports were selected, included, and excluded.

**Figure 2 ijerph-18-08941-f002:**
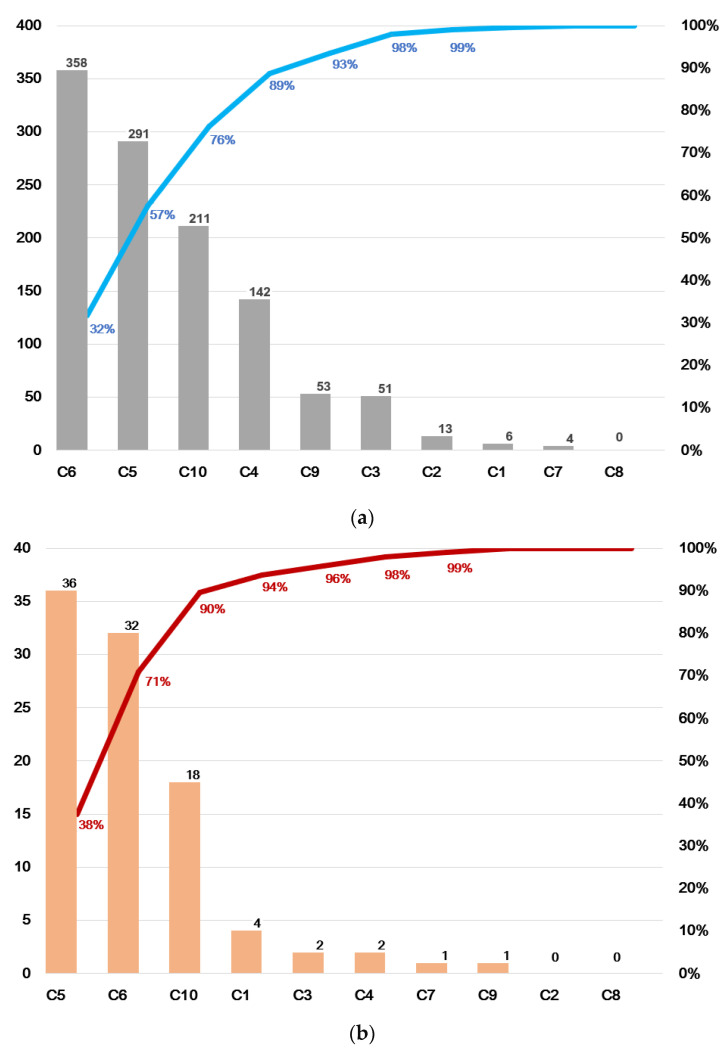
Pareto chart of the frequency of incidents according to WHO categorization. (**a**) Total PS incidents. (**b**) Adverse events. WHO 10-categories: C1: falls and other accidents; C2: patient behaviour; C3: clinical equipment and devices; C4: analogue and digital documentation; C5: clinical management and procedures; C6: clinical-administrative management; C7: infection associated with healthcare; C8: severe nosocomial pressure ulcers; C9: infrastructures and facilities, and C10: medication [30].

**Figure 3 ijerph-18-08941-f003:**
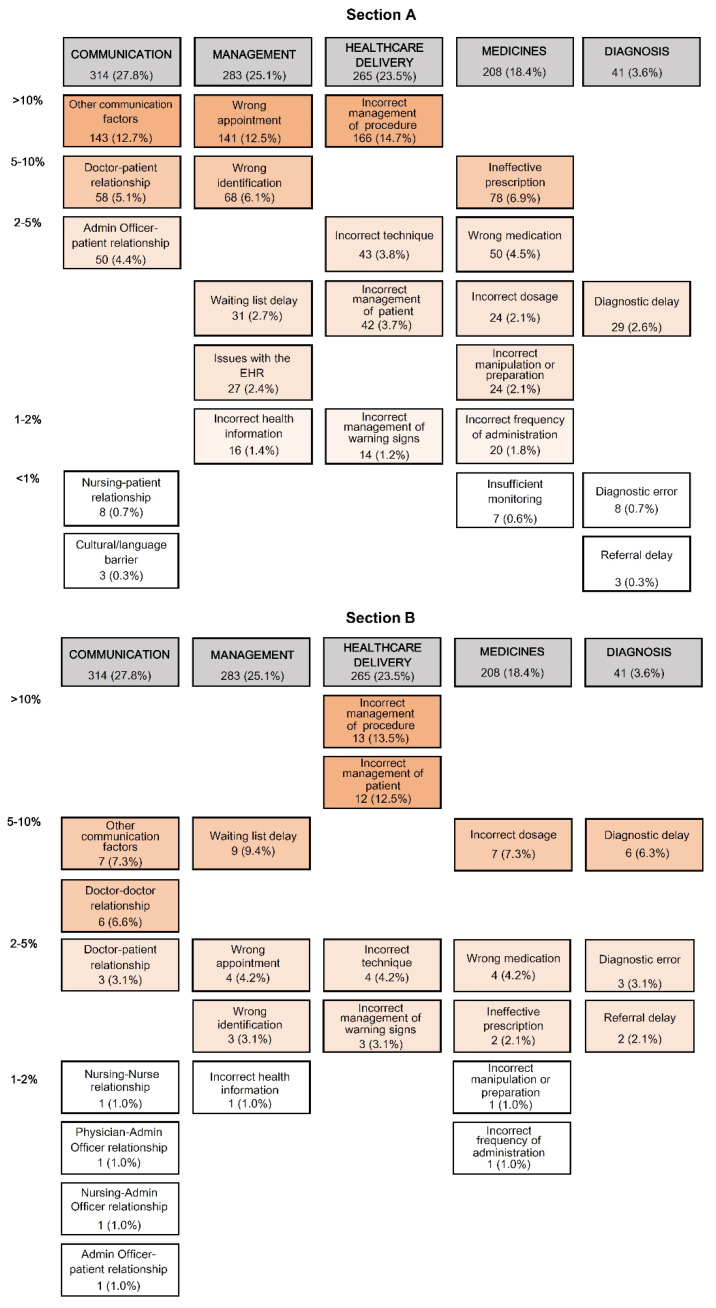
Risk map of the causal factors of the patient safety incidents. The figure shows the causal factors according to the APEAS model of the total patient safety incidents notified (section A; *n* = 1129) and adverse events notified (section B; *n* = 96) [19].

**Figure 4 ijerph-18-08941-f004:**
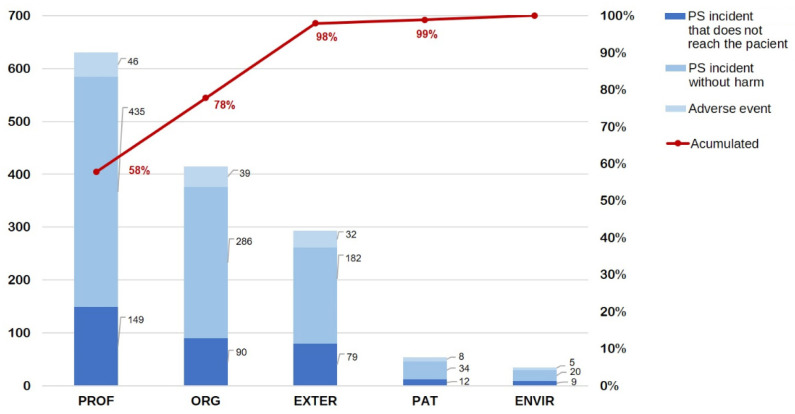
Pareto diagram of contributing factors frequencies in relation to the severity of the patient safety incident. The figure shows the contributing factors according to the WHO classification of the total patient safety incidents notified (*n* = 1129) [7]. Each incident can have one or more contributing factors. PROF: professional, ORG: organisation; EXTER: external; PAT: patient; ENVIR: environment.

**Table 1 ijerph-18-08941-t001:** Distribution of type of incidents notified according to the severity and clinical repercussions.

	Total PS Incidents	PS Incident Not Reach the Patient	PS Incident without Harm	Adverse Event
Incidents notified	1129 (100.0)	282 (25.0)	751 (66.5)	96 (8.5)
Severity of incident and clinical repercussions *				
Circumstance that might cause error	103 (9.1)	95 (33.7)	8 (7.8)	
Error has occurred but has been detected before reaching the patient	193 (17.1)	177 (62.8)	16 (8.3)	
Error has occurred without causing harm	733 (64.9)	7 (2.5)	726 (99.0)	
Observation required; no harm caused	49 (4.3)	3 (1.1)	1 (2.0)	45 (46.9)
Treatment required and/or temporary harm	33 (2.9)			33 (34.4)
Temporary damage has been caused that has required or lengthened hospitalization	13 (1.2)			13 (13.5)
Permanent damage has occurred	3 (0.3)			3 (3.1)
A near-death situation has occurred	2 (0.2)			2 (2.1)

Values are shown as absolute and percentage for each type of incident: total patient safety (PS) incident, PS incident that does not reach the patient (notifiable and near-miss circumstances), PS incident without harm and adverse event. (*) *p* value < 0.01.

**Table 2 ijerph-18-08941-t002:** Distribution of type of incidents notified according to probability and impact or severity of damage (risk matrix).

		PROBABILITY				
		Very Rare	Uncommon	Possible	Probable	Frequent
**IMPACT/SEVERITY**	**Does not reach patient: notifiable circumstances**	6 (0.5)	11 (1)	24 (2.1)	20 (1.8)	41 (3.6)
	**Does not reach patient: near-miss circumstances**	17 (1.5)	24 (2.1)	44 (3.9)	47 (4.2)	59 (5.2)
	**Minimal**	33 (2.9)	79 (7)	140 (12.4)	170 (15.1)	297 (26.3)
	**Minor**	3 (0.3)	3 (0.3)	16 (1.4)	7 (0.6)	19 (1.7)
	**Moderate 1**	2 (0.2)	4 (0.4)	9 (0.8)	11 (1)	4 (0.4)
	**Moderate 2**	2 (0.2)	2 (0.2)	4 (0.4)	4 (0.4)	1 (0.1)
	**Critical 1**		2 (0.2)			1 (0.1)
	**Critical 2**		1 (0.1)	1 (0.1)		

Values are shown as absolute and percentage for each type of incident. Colour legend; very low risk (cyan), low risk (green), moderate risk (yellow), high risk (salmon), and extreme risk (red).

**Table 3 ijerph-18-08941-t003:** Distribution of type of incidents notified categorised according to the Catalan Department of Health model.

	Total PS Incidents	PS Incident Not Reach the Patient	PS Incident without Harm	Adverse Event
Incidents notified	1129 (100.0)	282 (25.0)	751 (66.5)	96 (8.5)
Type of incident notified (related to) categorised according to the Department of Health model *				
Administrative processes	259 (22.9)	48 (17.0)	201 (26.8)	10 (10.4)
Lab	244 (21.6)	53 (18.8)	183 (24.4)	8 (8.3)
Safe use of medicines	184 (16.3)	62 (22.0)	104 (13.8)	18 (18.8)
Continuity of care	77 (6.8)	9 (3.2)	55 (7.3)	13 (13.5)
General services	69 (6.1)	27 (9.6)	41(5.5)	1 (1.0)
Diagnostic imaging	60 (5.3)	6 (2.1)	38 (5.1)	16 (16.7)
Healthcare process	60 (5.3)	13 (4.6)	40 (5.3)	7 (7.3)
Emergency care	50 (4.4)	7 (2.5)	18 (2.4)	
Vaccines	46 (4.1)	24 (8.5)	22 (2.9)	
Management of clinical material	22 (1.9)	14 (5.0)	8 (1.1)	
Ethics and rights of citizens	17 (1.5)	2 (0.7)	14 (1.9)	1 (1.0)
Infection surveillance, prevention and control	3 (0.3)		1 (0.1)	2 (2.1)
Waste management	2 (0.2)		2 (0.3)	
Health education	1 (0.1)		1 (0.1)	

Values are shown as absolute and percentage for each type of incident: total patient safety (PS) incident, PS incident that does not reach the patient (notifiable and near-miss circumstances), PS incident without harm and adverse event. (*) *p*-value < 0.01.

**Table 4 ijerph-18-08941-t004:** Distribution of the type of incidents reported according to the resolution level.

	Total PS Incidents	PS Incident Not Reach the Patient	PS Incident without Harm	Adverse Event
Incidents notified	1129 (100.0)	282 (25.0)	751 (66.5)	96 (8.5)
Resolution level *				
Health Centre	528 (46.8)	166 (58.9)	320 (42.6)	42 (43.8)
Patient Safety Functional Unit	211 (18.7)	20 (7.1)	156 (20.8)	35 (36.5)
Primary Care Management	312 (27.6)	62 (22.0)	236 (31.4)	14 (14.6)
Other (ICS, CatSalut.…)	78 (6.9)	34 (12.1)	39 (5.2)	5 (5.2)

Values are shown as absolute and percentage for each type of incident: total patient safety (PS) incident, PS incident that does not reach the patient (notifiable and near-miss circumstances), PS incident without harm and adverse event. (*) *p*-value < 0.01.

**Table 5 ijerph-18-08941-t005:** Distribution of type of incidents notified based on improvement actions.

	Total PS Incidents	PS Incident Not Reach the Patient	PS Incident without Harm	Adverse Event
Incidents notified	1129 (100.0)	282 (25.0)	751 (66.5)	96 (8.5)
Improvement actions in the same health centre * (*n* = 604; 53.5%)				
Committee/management	218 (19.3)	49 (17.4)	142 (18.9)	27 (28.1)
Training	204 (18.1)	58 (20.6)	197 (26.2)	36 (37.5)
Report review	166 (12.6)	63 (22.3)	127 (16.9)	14 (14.6)
Improvement team	16 (1.4)	2 (0.7)	14 (1.9)	
Improvement actions in the Patient Safety Functional Unit * (*n* = 525; 46.5%)				
Committee/management	93 (8.2)	21 (7.4)	48 (6.4)	24 (25.0)
Improvement team	21 (1.9)		16 (2.1)	15 (15.6)
Report review	6 (0.5)		6 (0.8)	
Warning	1 (0.1)			1 (1.0)

Values are shown as absolute and percentage for each type of incident: total patient safety (PS) incident, PS incident that does not reach the patient (notifiable and near-miss circumstances), PS incident without harm and adverse event. (*) *p*-value < 0.01.

## Data Availability

The data of the study have been obtained directly from the reports stored in the TPSC Cloud™ application of the corporate intranet of the Tarragona Regional Management of the Catalan Institute of Health. All records are anonymised. All data are considered confidential and treated according to Regulation 2016/679 of the European Parliament and Council of 27 April 2016 on Data Protection and Organic Law 3/2018, of 5 December, on the protection of personal data and guarantee of digital rights. Access is restricted to the research team by password. Data are available on resasonable request. The full dataset and statistical code are available from the first author (principal investigator) mgens.tgn.ics@gencat.cat.

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
