# Peer review of "Analysis of Patient Safety Incidents in Primary Care Reported in an Electronic Registry Application"

_ijerph, 2021, doi:10.3390/ijerph18178941_

Round 1

Reviewer 1 Report

This article analyzed the PS incidents and characters at PHCs in Spain, and provided valuable information to improve PS. The study was performed properly and the manuscript was well written. I only required authors to provide more background information to help audience to understand the study settings, samples, and potential bias caused  by the voluntary reporting. 

ABSTRACT is clearly written, with sufficient information about study design and findings. 

INTRODUCTION is also well organized with great background information about PS and current gap of PS from PHC. If possible, could you please add a short brief about the PHC in Spain? What types of services are provided, etc. This will help the audience understand the PS incidents better.

Methods: Since reporting to the PS system is voluntary (but encouraged), is there any information about the reporting rate? generally, severe events might be reported comparing to minor ones, and certain types of service providers might be more likely to report than others, etc. 

is it possible to also report the total amount of services in the selected PHCs during the study period? you can use it to have the PS reporting  rate/service amount, and form some comparison with other studies. 

Author Response

Thank you for your comments.

With regard to the introduction, the information on PHC in Spain can be found in the “setting” section. The paragraph is marked in blue in the manuscript.

“The Spanish primary care system differs from that of other countries in the European Community and the United States. The Spanish National Health System provides universal health coverage, is divided into 18 autonomous health systems and organised in health regions. In the Catalan Health Service, each health region has several primary care centres and at least one referral hospital. The health centres are managed with public funds and owned by the regional health authority. Each primary care centre has a multidisciplinary team that consists of family doctors, paediatricians, nurses, dentists, physiotherapists, psychologists, technical assistants and administrative staff. The centres provide general care in the community to patients of all ages, who attend appointments and/or urgent consultations. Health education and promotion, elderly care, adult care, mental healthcare, children's health, women's health and reproductive health are included in the services offered by primary care centres. In general, there is a high provision of these services [22]. All visits are recorded with the use of electronic medical records integrated into a clinical workstation (eCAP), which also supports medication prescription, which allows certain medication errors to be detected online [23]”.

We also believe that severe events are more reported than minor adverse effects, and that certain specialists are more likely to report them. We explain this in the Discussion (Summary of main findings section, lines 440-444). Paragraph marked in blue in the manuscript .

The most common notifications relate to incidents that reached the patient (75%), although only a small percentage caused adverse events. The adverse events reported were usually mild, and only a small fraction (0.5%) resulted in permanent damage or were life threatening. A minority of incidents (3.5%) were categorised as high or extreme risk, and most (99.2%) were considered preventable.

We have also calculated the notification rate based on the total visits provided during the study period in the participating primary care services. We have added these data in the results section (lines 285-294). We also discuss this aspect in the section “Comparison with existing literature” (lines 452-454).

Reviewer 2 Report

The article is very interesting and concerns a current issue: that of incident reporting in health facilities. The strong point is the geographical distribution of the data received and the fact that they concern the Emergency Department which, as per the literature, is one of the hospitals with the highest incidence of adverse events to the detriment of patients and healthcare operators.

There are two weaknesses that could be implemented: 1) Data on incident reporting systems are already known in the literature, on a larger scale. This magazine also dealt with the topic. Therefore, in the paragraph "Comparison with existing literature" you can also evaluate other known experiences (eg: https://doi.org/10.3390/ijerph17176267; but also doi: 10.1097 / MD.0000000000012509.). 2) Incident reporting is a tool that forms the basis of a series of clinical risk management activities which subsequently serve to reduce the occurrence of adverse events. You need to better explain this concept in the conclusions to also give a perspective view of the article, while in the discussions you could introduce this topic by enriching the bibliographic references and citing clinical risk management tools that serve to reduce adverse events after they have been detected. through incident reporting (eg Patient Safety Walkaround, Handover checklist, etc).

Author Response

Thank you for your comments.

As suggested, we have evaluated and added other examples. You can see the new text in green in the discussion section (lines 568-575).

We also believe that “incident reporting is a tool that forms the basis of a series of clinical risk management activities which subsequently serve to reduce the occurrence of adverse events”. We refer to this concept in the section “Implications for clinical practice and future research”. We have also added this concept in the conclusions (lines 686-689).

Reviewer 3 Report

Please see attached

Author Response

Thank you for your comments.

We also believe that primary care patient safety varies in different regions of the world. We have thus added the following paragraph “PS in primary care cannot be considered universal, each region needs to carry out its own specific research” (620-621). We have also added the expression “depending on the level of analysis” (“Comparison with existing literature”, line 516-517). Marked in yellow  in the manuscript.

It is accepted that PS incidents and adverse events are usually replicated in similar institutions with similar causes and contributing factors, depending on the level of analysis [41].

We have changed “physician” by “doctor” in the manuscript, since in most cases we refer to “qualified specialists in family medicine”. This concept also includes doctors working in the community rather than at health care facilities. All these specialists have been included in the analysis.

In the section “Reporting System”, we mention different tools commonly used in  PS management (risk matrix, cause-effect, etc).  We have briefly explained these tools to clarify their use in this study (lines 149-161).

In relation with the definition of safety incident and adverse event (section “Taxonomy and definitions”), we will follow your suggestion to consider an incident as a “release of something (more or less harmful)”, in agreement with the current literature on patient safety. We have reviewed and changed the definitions of  safety incident and adverse event to a simplified version also accessible to the non-medical readership. Marked in the manuscript in yellow (section “Taxonomy and definitions“, lines 210-212). We use the terminology of the WHO Conceptual Framework for the International Classification of Patient Safety (http://www.who.int/patientsafety/taxonomy/icps_full_report.pdf). The concept “unnecessary” refers to the preventable harm and compares with necessary situations such as the incision for a laparotomy, which are not considered incidents. However, we have chosen not to use “unnecessary harm’ for the data and analyses for this study.

Another issue clarified in the section “Notification evaluation procedure” is the role of the experts at the  Central Safety Functional Unit of the Primary Care Management: clinicians, pharmacologists, nurses, administrative officers and statisticians (lines 238-247). While their regular training does not make them patient safety experts, they all have received postgraduate academic training in patient safety, they are all experienced and have a leading role in this essential area of health care.

In the paragraph of the section “Statistical Analysis“, we clarify that “these priority areas” refer to the critical PS areas that the experts at Central Safety Functional Unit of the Primary Care Management considered a priority (line 269-271).

In the “Results” section, we explain that all notification reviews were done by the patient safety experts mentioned earlier in the article (lines 277).

Also in “Results”, we add further data on notifications in relation to the number of health services provided in those particular centres (lines 291-294). Results show that the difference between reporting by primary care centres on the one hand, and sexual and reproductive healthcare centres and emergency care centres on the other, is not attributable to the larger volume of patient-staff interactions, since the differences persist after considering number of patient-professional encounters. This aspect is later discussed in the section “Comparison with existing literature” (lines 452-454).

In the section “Causal and contributing factors”, we have removed “in the global analysis” and added “In all reported incidents,…” (line 356). Additionally, in Figure 4 “Pareto diagram of contributing factors frequencies in relation to the severity of the patient safety incident”, we explain that each incident can have one or more contributing factors. In contrast, in the results shown in Figure 3 “Risk map of the causal factors of the patient safety incidents”, each incident only has a causal factor. We explain that Figure 3 is independent of Figure 4.

In the section “Resolution level and proposed actions”, specifically in the paragraph starting “The main proposals were”, we explain that “95% of the professionals in primary care centres have received PS training”. This training is conducted yearly and consists of 6-hour workshops accredited by the Institut Català de la Salut (lines 405-406).

Also in this section, as suggested, to help readers understand second and third victim concepts, we have added an additional paragraph with a literature reference (we are aware that citations are uncommon in the results section).  Marked in yellow  in the manuscript (lines 408-418).

In the discussion section, as suggested, we have clarified the analysis/results related to incident prevention (see lines 555-556). We have also added a new reference [Sujan, M. A., Huang, H., & Braithwaite, J. (2017). Learning from incidents in health care: Critique from a Safety-II perspective. Safety Science, 99, 115-121], which complements references [13, 52](line 575-576).

Finally, in table S1 of the supplementary material, we have added a paragraph on the quality categories of incident notification (correct and discrepant).
